# Financial Toxicity among Patients with Breast Cancer during the COVID-19 Pandemic in the United States

**DOI:** 10.3390/cancers16010062

**Published:** 2023-12-21

**Authors:** Yan Wu, Xianchen Liu, Martine C. Maculaitis, Benjamin Li, Alexandra Berk, Angelina Massa, Marisa C. Weiss, Lynn McRoy

**Affiliations:** 1Ernest Mario School of Pharmacy, Rutgers University, Piscataway, NJ 07103, USA; yan.wu@pfizer.com; 2Pfizer Inc., New York, NY 10001, USA; benjamin.li@pfizer.com (B.L.); lynn.mcroy@pfizer.com (L.M.); 3Oracle Life Sciences, Austin, TX 78741, USA; 4Invitae Corporation, San Francisco, CA 94103, USA; alexaberk@gmail.com (A.B.); angelina.massa@invitae.com (A.M.); 5Breastcancer.org, Ardmore, PA 19003, USA; mweiss@breastcancer.org

**Keywords:** breast cancer, COmprehensive Score for financial Toxicity (COST), COVID-19 Stress Scale, COVID-19, financial toxicity

## Abstract

**Simple Summary:**

Financial toxicity, defined as the financial distress experienced by patients related to their cancer care, can adversely impact clinical outcomes. Little is known about financial toxicity and associated factors among patients with breast cancer during the COVID-19 pandemic. Our study of 669 patients shows that financial toxicity was prevalent in patients with breast cancer, particularly those with metastatic disease, during COVID-19. Lower income and elevated depression symptom severity were the primary factors associated with greater financial toxicity and socioeconomic stress from COVID-19. Patient financial burden is substantial and must be considered when managing breast cancer.

**Abstract:**

This study reported the prevalence of financial distress (financial toxicity (FT)) and COVID-19-related economic stress in patients with breast cancer (BC). Patients with BC were recruited from the Ciitizen platform, Breastcancer.org, and patient advocacy groups between 30 March and 6 July 2021. FT was assessed with the COmprehensive Score for financial Toxicity (COST) instrument. COVID-19-related economic stress was assessed with the COVID-19 Stress Scale. Among the 669 patients, the mean age was 51.6 years; 9.4% reported a COVID-19 diagnosis. The prevalence rates of mild and moderate/severe FT were 36.8% and 22.4%, respectively. FT was more prevalent in patients with metastatic versus early BC (*p* < 0.001). The factors associated with FT included income ≤ USD 49,999 (adjusted odds ratio (adj OR) 6.271, *p* < 0.0001) and USD 50,000–USD 149,999 (adj OR 2.722, *p* < 0.0001); aged <50 years (adj OR 3.061, *p* = 0.0012) and 50–64 years (adj OR 3.444, *p* = 0.0002); living alone (adj OR 1.603, *p* = 0.0476); and greater depression severity (adj OR 1.155, *p* < 0.0001). Black patients (adj OR 2.165, *p* = 0.0133), patients with income ≤ USD 49,999 (adj OR 1.921, *p* = 0.0432), or greater depression severity (adj OR 1.090, *p* < 0.0001) were more likely to experience COVID-19-related economic stress. FT was common in patients with BC, particularly metastatic disease, during COVID-19. Multiple factors, especially lower income and greater depression severity were associated with financial difficulties during COVID-19.

## 1. Introduction

Treatment for cancer poses a major financial burden to patients and is projected to increase [1]. Patients with breast cancer (BC) are at particular risk of financial hardship, given the protracted nature of BC and the need for longitudinal multidisciplinary care [2]. Specifically, care for female patients with BC had the highest national expenditure among all cancer types in the United States (US) at USD 29.8 billion in 2020 [1]. Additionally, total out-of-pocket expenses for BC care were estimated to be the highest among all cancers in 2019 at USD 3.1 billion [3]. Further, total mean costs per patient were USD 140,732 for the first year of care in 2016 for privately insured adult patients with BC, and mean per patient out-of-pocket costs were USD 6000 [4].

Financial toxicity (FT), defined as the financial distress experienced by patients related to cancer treatment, is increasingly recognized as an adverse outcome of cancer care [5]. FT encompasses the objective financial consequences of treatment costs and the subjective financial concerns. The “toxicity” of the cancer-related financial burden can manifest in the form of multiple consequences for patients. For example, a recent systematic review that analyzed the global burden of FT specifically among patients with BC showed heterogeneity in the definition of FT, ranging from trouble paying for necessities like food and medical bills, going into debt, or even having to forgo medical treatment altogether [2]. Patients with lower incomes are particularly susceptible to financial hardships, as poverty can be a major barrier to accessing care, which in turn, may result in later-stage cancer diagnoses that require extensive treatment interventions [2]. The association between employment status and FT has also been characterized, with prior research showing that cancer diagnoses negatively affect the individual’s ability to maintain employment, which can lead to or exacerbate FT [6,7]. Adverse health-related outcomes are also associated with FT. Two recent systematic reviews on the impact of FT on patients with cancer reported an association between higher FT and worse health-related quality of life in both the physical and mental health domains from several patient-reported outcome (PRO) instruments [8,9].

The COVID-19 pandemic has exacerbated financial stress among patients with cancer, including increased debt and difficulty in paying for both health care and non-health care needs [10,11]. The pandemic also resulted in the loss of employment or furlough [11], and pandemic-related distress was associated with maladaptive behaviors, including substance abuse, panic buying, and overspending [12,13]. In patients with BC, the COVID-19 pandemic disrupted cancer care in several ways. Cancer-screening rates were reduced by 47% [14], whereas diagnoses declined by as much as 29% between 2019 and 2021 [15]. Cancer-related mortality increased in the first year of the pandemic (2020), compared with 2019, and has been attributed to increased vulnerability to COVID-19 and related complications in patients with cancer [16]. Pandemic-related disruption to BC screening was projected to increase the risk of death and hospitalization in patients with BC, attributable to advanced stage at diagnosis and delayed treatment [17,18,19]. Advanced disease was also associated with higher chemotherapy costs during the pandemic, possibly owing to a higher demand for treating patients with stage III–IV disease [20].

While there is an increasing body of research exploring FT and its impact on patients, data on FT among patients with BC during the COVID-19 pandemic are limited. Accordingly, the current study sought to address this information gap by reporting the prevalence of FT and COVID-19-related economic stress in patients with BC. This study further expanded upon our previous research, which assessed the factors contributing to the psychosocial well-being of patients with BC during the COVID-19 pandemic [21], by evaluating FT, socioeconomic stress, and the associated demographic and clinical characteristics among patients with BC during the COVID-19 pandemic. 

## 2. Methods

### 2.1. Study Design and Data Collection

This cross-sectional study collected data from patients with BC via an online self-report survey conducted between 30 March and 6 July 2021 [21]. Patients with BC were recruited from the Ciitizen platform (now part of Invitae Corporation, San Francisco, CA, USA), Breastcancer.org, and patient advocacy groups (METAvivor, TOUCH, The Breasties, SurvivingBreastCancer.org, and the Metaplastic Breast Cancer Global Alliance). The Ciitizen platform collects and stores patient medical records on patients’ behalf in compliance with the Health Insurance Portability and Accountability Act (HIPAA) right of access. Medical record documents are used to generate structured, longitudinal data that can be shared for the patient’s own clinical treatment, for observational research, or clinical trials. Informed consent was obtained from all patients involved in the study. Key clinical and treatment data were extracted from the medical records of patients who had consented to participate and share their de-identified data. Approximately 39% of participants were recruited from the Breastcancer.org patient community. 

### 2.2. Institutional Review Board Statement

The study protocol was reviewed by Pearl IRB (Indianapolis, IN, USA) (IRB Protocol #: 21-CITI-121). The IRB approved the study and determined that the study was exempt according to FDA CFR §56.104 and 45 CFR §46.104(b) (2, 4): (2) Tests, Surveys, Interviews; (4) Secondary Research Uses of Data, on 30 March 2021. Participants gave informed written consent to participate in this study, which only used de-identified patient data. The study was carried out in accordance with the Declaration of Helsinki.

### 2.3. Patient Selection

This study included patients aged ≥18 years who were diagnosed with early BC (eBC, stage I, II, or III) or metastatic BC (mBC, stage IV) by a physician at the time of survey. Patients who had stage 0 BC or ductal carcinoma in situ (DCIS) at the time of the survey were excluded. All patients provided informed consent and were able to complete the survey. 

### 2.4. Study Measures

Data on sociodemographic and clinical characteristics, FT, and history of COVID-19 diagnosis were collected via an online self-reported survey. The Charlson Comorbidity Index (CCI) score was computed by summing weights assigned to 11 health conditions, with higher scores indicating greater comorbidity burden [22]. The 8-Item Patient Health Questionnaire (PHQ-8) was used to measure depression; scores range from 0 to 24, with scores ≥ 10 indicating potential depression [23]. 

FT was assessed with the COomprehensive Score for financial Toxicity (COST) instrument. The COST, a validated questionnaire, includes 11 items [24]. Each item is rated on a Likert scale from 0 (not at all) to 4 (very much). Patients rate questions pertaining to their financial situation, such as “I am able to meet my monthly expenses”, “I feel financially stressed”, and “My out-of-pocket medical expenses are more than I thought they would be”. A total scale score is computed by summing the scores on the 11 items, ranging from 0 to 44. Lower COST scores indicate higher levels of FT. The cut-offs applied for prevalence of FT were as follows: COST scores ≥ 26 indicate no FT; COST scores of 14–25 indicate mild FT; COST scores of 0–13 indicate moderate/severe FT [24]. Cronbach’s alpha was 0.90 for COST scale with the current sample, indicating high internal consistency among the scale items.

The COVID-19 Stress Scale (CSS) was used to measure different aspects of COVID-19-related stress [25]. The CSS is a 36-item scale, divided into 6 subscales: danger, socioeconomic consequences, xenophobia, contamination, traumatic stress, and compulsive checking. In this study, only the CSS Socioeconomic (CSS-SE) subscale was used as a tool to measure the socioeconomic stress due to the COVID-19 pandemic. The CSS-SE scale consists of 6 items. Examples of items assessed in the CSS-SE scale include “I am worried about grocery stores running out of food”, “I am worried that grocery stores will close down”, and “I am worried about pharmacies running out of prescription medicines”. Each item is rated on a 5-point Likert scale from 0 (not at all) to 4 (extremely). The CSS-SE scores range from 0 to 24. Higher scores indicate greater stress from socioeconomic consequences during the COVID-19 pandemic. Patients were divided into 2 groups based on the median CSS-SE score from this study: CSS-SE scores > 2 indicate high socioeconomic stress and CSS-SE scores ≤ 2 indicate low socioeconomic stress. Cronbach’s alpha was 0.92 for the CSS-SE scale with the current sample, indicating high internal consistency among the scale items. 

### 2.5. Statistical Analyses

Descriptive statistics were reported as frequencies and percentages for categorical variables, and mean and standard deviation (SD) for continuous variables. Comparisons of mean COST scores and CSS-SE scores and prevalence rates of FT by BC type and COVID-19-diagnosis history were performed using two-sample *t*-tests and chi-square (c^2^) tests, respectively. Multivariable logistic regression was used to determine significant factors associated with FT (COST scores < 26: mild and moderate/severe) and high socioeconomic stress (CSS-SE scores > 2) from the COVID-19 pandemic. Odds ratio (OR) with 95% confidence intervals (CIs) were used to show the association of variables with FT. For all statistical analyses, a *p* value of < 0.05 (2-tailed) was considered statistically significant. Analyses were performed using SAS version 9.4 (Cary, NC, USA).

## 3. Results 

### 3.1. Patient Baseline Demographics and Clinical Characteristics

A total of 669 patients were included in the study. The mean age was 51.6 years (range: 28–82 years); 561 patients (83.9%) were White, 662 (99.0%) were female, and 344 (51.4%) had mBC when completing the survey. A total of 62 patients (9.4%) reported a previous diagnosis with COVID-19. Endocrine therapy was the most common systemic treatment for BC (58.7%), followed by targeted therapy (38.4%) and chemotherapy (21.4%). Most patients (68.9%) were married or living with a partner, approximately half (51.0%) were employed, 69.5% had a college degree or higher, and 62.0% had commercial health insurance. Over a quarter of patients (26.0%) earned an annual income < USD 50,000, and 28.7% had out-of-pocket monthly costs exceeding USD 300. Table 1 provides detailed patient characteristics.

### 3.2. COST Scores

The mean COST score (± SD) for the total study sample was 22.7 ± 10.8 (Figure 1). Mean COST scores were significantly higher in patients with eBC than in those with mBC (24.2 ± 11.3 vs. 21.3 ± 10.2, t = 3.458, *p* < 0.001), indicating higher levels of FT in patients with mBC than in those with eBC. Mean COST scores did not differ significantly by COVID-19-diagnosis history (21.3 ± 9.8 vs. 22.9 ± 10.9, t = 1.121 *p* = 0.263).

### 3.3. Prevalence of FT

In the sample, 273 patients (40.8%) did not experience FT (COST scores ≥ 26), 246 (36.8%) experienced mild FT (COST scores 14–25), and 150 (22.4%) experienced moderate/severe FT (COST scores 0–13). The prevalence of FT was significantly higher in patients with mBC (no FT: 33.7%, mild FT: 42.2%, moderate/severe FT: 24.1%), compared with patients with eBC (no FT: 48.3%, mild FT: 31.1%, moderate/severe FT: 20.6%; χ^2^ = 15.2067, *p* < 0.001). No statistically significant difference (χ^2^= 0.6690, *p* = 0.7157) in the prevalence of FT was observed between patients with and without a COVID-19-diagnosis history (Figure 2).

Multivariable logistic regression analysis was performed to examine factors associated with greater FT (COST scores < 26) (Table 2). Lower income was significantly associated with greater FT when comparing income ≤ USD 49,999 vs. ≥ USD 150,000 (adjusted (adj) OR = 6.271; 95% CI: 2.992–13.142, *p* < 0.0001), and income USD 50,000 to USD 149,999 vs. ≥ USD 150,000 (adj OR = 2.722; 95% CI: 1.678–4.416, *p* < 0.0001). Younger age was also significantly associated with greater FT; aged < 50 years vs. ≥ 65 years (adj OR = 3.061; 95% Cl: 1.557–6.015, *p* = 0.0012) and 50–64 years vs. ≥ 65 years (adj OR = 3.444; 95% CI: 1.795–6.609, *p* = 0.0002). Elevated PHQ-8 score was significantly associated with greater FT (adj OR = 1.155; 95% CI: 1.106–1.207, *p* < 0.0001). Not married/living with partner (category “other”) was associated with greater FT vs. married/living with partner (adj OR = 1.603; 95% CI: 1.005–2.556, *p* = 0.0476). However, lower out-of-pocket costs for BC treatment were associated with significantly less FT for out-of-pocket costs of USD 0 (adj OR = 0.171; 95%CI: 0.081–0.361, *p* < 0.0001) and out-of-pocket costs of USD 1 to USD 100 (adj OR = 0.458; 95%CI: 0.246–0.852, *p* = 0.0137), compared with out-of-pocket costs of USD 501 or more.

### 3.4. Socioeconomic Stress from the COVID-19 Pandemic

The mean CSS-SE score (± SD) for the total study sample was 3.7 ± 4.8. The mean CSS-SE score did not differ significantly by BC type (eBC: 3.5 ± 4.8 vs. mBC: 3.9 ± 4.9; *p* = 0.339) or by COVID-19-diagnosis history (yes: 4.6 ± 5.7 vs. no: 3.6 ± 4.7; *p* = 0.122). 

The median CSS-SE score for the sample was 2.0. Multivariable logistic regression was performed to examine factors associated with high socioeconomic stress due to the COVID-19 pandemic (i.e., CSS-SE > median score) (Table 3). The multivariable analysis showed that non-White race was associated with more socioeconomic stress due to the pandemic (Black vs. White (adj OR = 2.165; 95% CI: 1.175–3.990, *p* = 0.0133), and Other race vs. White (adj OR = 1.938; 95% CI, 1.005–3.736, *p* = 0.0481)), although both non-White groups had relatively small sample sizes. Lower income was also associated with higher socioeconomic stress due to the pandemic; income ≤ USD 49,999 vs. ≥ USD 150,000 (adj OR = 1.921; 95% CI: 1.020–3.617, *p* = 0.0432). Elevated PHQ-8 score was another significant factor that was associated with more socioeconomic stress due to the pandemic (adj OR = 1.090; 95% CI: 1.051–1.130, *p* < 0.0001).

## 4. Discussion

The term “financial toxicity” was first applied to describe the adverse financial effects stemming from medical treatment related to cancer in general [5]. Since then, several studies have aimed to identify populations vulnerable to FT in more specific types of cancer, such as BC [2,26,27], which was the most expensive cancer to treat in the US in 2020 [1]. Globally, FT is highly prevalent among patients with BC, with differing rates according to country income levels [2]. Two recent systematic reviews reported that increased FT was associated with worse health-related quality of life for patients with cancer and reported an association between higher FT and greater mental health problems [8,9].

Limited data are available for FT associated with cancer care during the COVID-19 pandemic [11]. Our study is the first to examine FT and socioeconomic stress specifically among patients with BC during the pandemic, and shows that FT was common among patients with BC. Mild and moderate/severe FT was collectively experienced by 59.2% of the patients in this analysis. These results highlight the frequency of FT experienced by patients with BC in this cohort. Patients with mBC were more likely to have moderate/severe FT than patients with eBC. These findings are presumably a consequence of much higher costs for managing the advanced stages of BC than the early stages, including hospitalizations and treatments, which increase substantially over time. Lower income, younger age, and greater depressive symptom severity were independently associated with greater levels of FT. Higher income and older age have been associated with less FT in a recent study [28], whereas previous studies have reported the association between FT and depressive symptoms in both younger and older adults outside of the COVID-19 pandemic context [29,30].

A recent international meta-analysis showed that substantial FT (measured via COST score) was associated with BC treatment globally; FT was observed regardless of patients’ income level, although patients with lower- and middle-income had higher rates of FT [2]. In another study of patients with BC who underwent surgery (lumpectomy or mastectomy), higher income and lower out-of-pocket costs were associated with lower FT [31]. This finding is consistent with those from the current study in which significantly higher FT was associated with incomes <USD 150,000 compared with ≥USD 150,000, and less FT was associated with lower (<USD 100 per month) out-of-pocket costs, compared with >USD 100 per month. The findings in this study are also consistent with previous studies reporting higher FT in younger, non-White women with BC with a lower socioeconomic status [32,33], and a cross-sectional study that reported patients with mBC were more susceptible to FT than those with eBC [34]. An additional study of women with eBC (stage 0–III BC) also showed an association of FT with age, income, and cancer stage [35]. In the present analysis, greater FT was also experienced by patients living alone than by patients who were living with a partner or who were married, likely the result of increased accessibility to shared financial resources in the latter group.

Our study additionally observed that a higher level of COVID-19-related socioeconomic distress (as measured by CSS-SE scores) was experienced by patients with lower income and elevated depression symptom severity (PHQ-8 scores). These associations are consistent with the negative impact of FT has on patients in the form of employment disruption, asset erosion, indebtedness, and psychological distress [36,37,38]. Of note, the multivariable analysis results suggested that non-White patients were more likely to experience socioeconomic stress during the COVID-19 pandemic. 

A number of previous studies corroborate these findings by demonstrating that individuals with lower income disproportionately faced economic hardship during the COVID-19 pandemic [39], as did Black individuals [40]. Racial and ethnic minority groups have previously been shown to have experienced more discrimination and inequities during the pandemic [41], which have been associated with psychological distress and sleep disturbances [39,40,41]. This study additionally shows that greater depression symptom severity was associated with FT and socioeconomic stress during the pandemic. Although further research is needed, the association between depression and financial difficulties may be linked to lifestyle disturbances during the pandemic associated with increased levels of depression [42,43,44], which in turn, lead to or exacerbate patient financial burden.

Potential future strategies to address FT include shared decision-making between patients and physicians; however, current data regarding the inclusion of FT in decision-making discussions are limited [45]. The COST tool can be used to identify patients who are at higher risk for financial burden, which could enable resources to be more effectively allocated to those patients who would most likely need and benefit from financial counseling, support, and education. The COST tool could provide an evidence-based approach to stimulate conversations between health care providers and patients who may be uncomfortable discussing their financial situation. Financial navigation, an assessment of a patient’s risk factors for FT and the coordination of access to financial assistance programs, which is most often led by cancer care support staff (e.g., nurses, social workers), may be utilized to improve clinical decision making for patients and lessen FT [46]. The implementation of financial counseling services at oncology care sites has the potential to additionally benefit patient psychosocial well-being [21,47,48], especially when coping with a public health crisis, such as the COVD-19 pandemic [47]. Prior research has shown that individuals with pre-existing anxiety and mood disorders reported higher CSS scores during the COVID-19 pandemic [49], which necessitates further study regarding how pandemic-related distress changes with counseling and medical intervention.

### Limitations

The following limitations should be considered when interpreting the findings. Patients who participated in the survey were not randomly selected. Accordingly, findings from these participants may not be generalizable to the broader population of patients with BC in the US. Furthermore, FT and socioeconomic stress during the COVID-19 pandemic may have been exacerbated by the worldwide public health crisis and the study findings may not generalize to patients with BC in less critical situations. However, as aforementioned, globally, FT is highly prevalent among patients with BC. Additionally, FT and socioeconomic stress were self-reported, which may be subject to recall bias. Finally, this is a cross-sectional study; therefore, causal relationships among study variables could not be inferred.

## 5. Conclusions

FT was common in patients with BC during the COVID-19 pandemic, particularly among those with mBC. Our study demonstrated that lower income and higher depression symptom severity were associated with greater FT and socioeconomic stress, while lower out-of-pocket costs and advancing age were associated with lower FT. These findings highlight the need to better understand the financial burden to improve the quality of life of patients with BC, especially for those with metastatic disease during a public health crisis. Although this study was conducted during the COVID-19 pandemic and the pandemic may have exerted a financial burden, the findings may have important implications to address FT beyond the COVID-19 pandemic as FT is prevalent and negatively impacts clinical outcomes and the psychosocial well-being of patients with breast cancer [2,6].

## Figures and Tables

**Figure 1 cancers-16-00062-f001:**
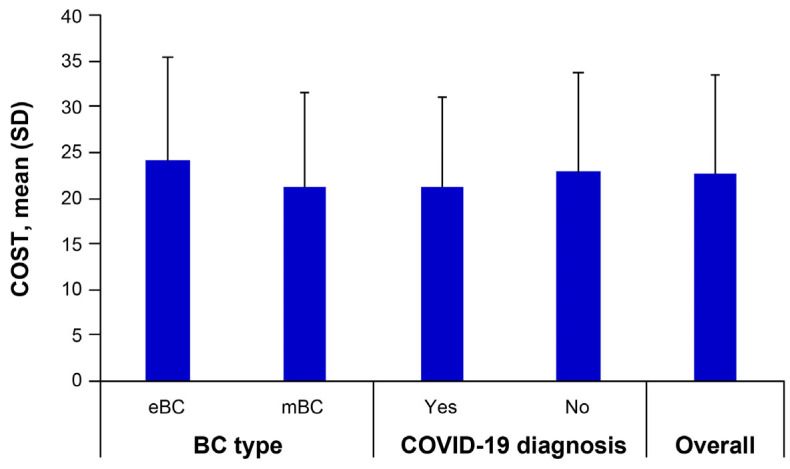
Mean COST scale scores by BC type and COVID-19 diagnosis. Error bars represent SD. BC, breast cancer; COVID-19, coronavirus disease 2019; COST, COmprehensive Score for financial Toxicity; eBC, early breast cancer; mBC, metastatic breast cancer; SD, standard deviation.

**Figure 2 cancers-16-00062-f002:**
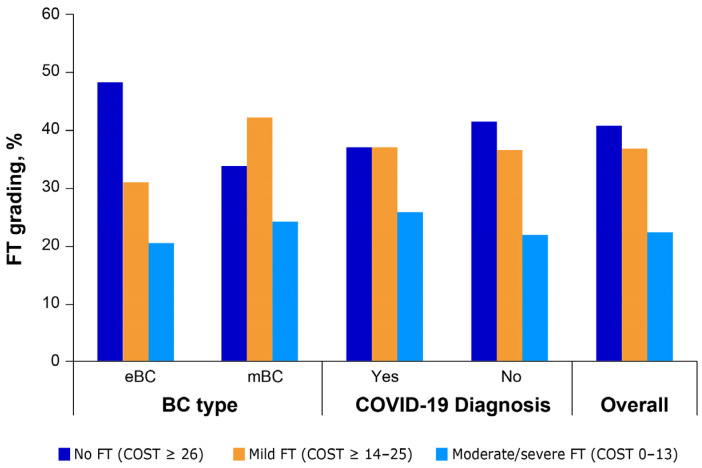
Prevalence of FT by BC type and COVID-19 diagnosis. BC, breast cancer; COVID-19, coronavirus disease 2019; COST, comprehensive score for financial toxicity; eBC, early breast cancer; FT, financial toxicity; mBC, metastatic breast cancer.

**Table 1 cancers-16-00062-t001:** Patient demographics and clinical characteristics.

Characteristic	Patients (*n* = 669)
Age, mean (SD), years	51.6 (11.3)
Gender	
Female, *n* (%)	662 (99.0)
Male, *n* (%)	7 (1.0)
Race, *n* (%) ^a^	
White	561 (83.9)
Black	60 (9.0)
Other	48 (7.2)
BC type, *n* (%)	
eBC	325 (48.6)
mBC	344 (51.4)
History of COVID-19 diagnosis, *n* (%)	
Yes	62 (9.4)
No	599 (90.6)
Current treatments, *n* (%)	
Endocrine therapy	393 (58.7)
Targeted therapy	257 (38.4)
Chemotherapy	143 (21.4)
Supportive/palliative care	90 (13.5)
Radiation	85 (12.7)
Surgery in past 3 months	74 (11.1)
Immunotherapy	23 (3.4)
Not receiving treatment	76 (11.4)
Marital status	
Married/living with partner	461 (68.9)
Other	208 (31.1)
Employment status	
Employed	341 (51.0)
Not employed	321 (48.0)
Prefer not to answer	7 (1.0)
Education status	
College degree or higher	465 (69.5)
Less than college degree	203 (30.3)
Prefer not to answer	1 (0.2)
Income	
≥USD 150,000	146 (21.8)
USD 50,000 to USD 149,999	321 (48.0)
≤USD 49,999	174 (26.0)
Prefer not to answer	28 (4.2)
Insurance	
Commercial	415 (62.0)
Medicare/Medicaid	165 (24.7)
Other	89 (13.3)
Region	
Rural/small town	238 (35.6)
Suburb/city	431 (64.4)
Out-of-pocket costs for BC in an average month	
USD 0	130 (19.4)
USD 1–USD 100	190 (28.4)
USD 101–USD 300	148 (22.1)
USD 301–USD 500	83 (12.4)
USD 501 or more	109 (16.3)
Do not know	9 (1.4)
BMI	
Obese	227 (33.9)
Overweight	179 (26.8)
Underweight/normal weight	263 (39.3)
CCI, mean (SD)	3.27 (2.40)
PHQ-8, mean (SD)	7.66 (5.26)

^a^ Values do not add up to 100% due to rounding. BC, breast cancer; BMI, body mass index; CCI, Charlson Comorbidity Index; COVID-19, coronavirus disease 2019; eBC, early breast cancer; mBC, metastatic breast cancer; PHQ-8, 8-item Patient Health Questionnaire; SD, standard deviation.

**Table 2 cancers-16-00062-t002:** Demographics and clinical characteristics associated with FT.

Variable	No FT: COST ≥ 26 *n* = 273	Yes FT: COST < 26 *n* = 396	Adj OR (95% CI)	*p* Value
Age, years				
<50	103 (37.73)	176 (44.44)	3.061 (1.557–6.015)	0.0012
50 to 64	114 (41.76)	186 (46.97)	3.444 (1.795–6.609)	0.0002
≥65	56 (20.51)	34 (8.59)	REF	
Sex				
Male	4 (1.47)	3 (0.76)	0.689 (0.105–4.510)	0.6979
Female	269 (98.53)	393 (99.24)	REF	
Race				
White	229 (83.88)	332 (83.84)	REF	
Black	22 (8.06)	38 (9.60)	1.097 (0.557–2.160)	0.7891
Other	22 (8.06)	26 (6.57)	0.738 (0.354–1.541)	0.4187
Marital status				
Married/living with partner	216 (79.12)	245 (61.87)	REF	
Other	57 (20.88)	151 (38.13)	1.603 (1.005–2.556)	0.0476
Employment status				
Employed	157 (57.51)	184 (46.46)	REF	
Not employed	113 (41.39)	208 (52.53)	0.899 (0.566–1.429)	0.6532
Prefer not to answer	3 (1.10)	4 (1.01)	0.827 (0.151–4.531)	0.8265
Education status				
College degree or higher	210 (76.92)	255 (64.39)	REF	
Less than college degree	63 (23.08)	140 (35.35)	1.169 (0.732–1.868)	0.5127
Income				
≤USD 49,999	35 (12.82)	139 (35.10)	6.271 (2.992–13.142)	<0.0001
USD 50,000 to USD 149,999	128 (46.89)	193 (48.74)	2.722 (1.678–4.416)	<0.0001
≥USD 150,000	92 (33.70)	54 (13.64)	REF	
Prefer not to answer	18 (6.59)	10 (2.53)	1.151 (0.427–3.101)	0.9863
Insurance				
Commercial	191 (69.96)	224 (56.57)	REF	
Medicare/Medicaid	51 (18.68)	114 (28.79)	1.708 (0.892–3.273)	0.1065
Other	31 (11.36)	58 (14.65)	1.299 (0.707–2.387)	0.3997
Region				
Rural/small town	104 (38.10)	134 (33.84)	0.735 (0.490–1.102)	0.1365
Suburb/city	169 (61.90)	262 (66.16)	REF	
Out-of-pocket costs for BC in an average month				
USD 0	74 (27.11)	56 (14.14)	0.171 (0.081–0.361)	<0.0001
USD 1–USD 100	88 (32.23)	102 (25.76)	0.458 (0.246–0.852)	0.0137
USD 101–USD 300	50 (18.32)	98 (24.75)	0.825 (0.441–1.542)	0.5463
USD 301–USD 500	26 (9.52)	57 (14.39)	0.866 (0.416–1.804)	0.7015
USD 501 or more	33 (12.09)	76 (19.19)	REF	
Do not know	2 (0.73)	7 (1.77)	1.215 (0.193–7.633)	0.8358
BC type				
eBC	157 (57.51)	168 (42.42)	REF	
mBC	116 (42.49)	228 (57.58)	1.091 (0.626–1.899)	0.7587
COVID-19 diagnosis				
No	249 (91.21)	350 (88.38)	0.969 (0.504–1.859)	0.2069
Not answered	1 (0.37)	7 (1.77)	4.358 (0.400– 47.417)	0.2268
Yes	23 (8.42)	39 (9.85)	REF	
BMI				
Underweight/normal weight (BMI < 25 kg/m^2^)	121 (44.32)	142 (35.86)	REF	
Overweight (BMI ≥ 25 to < 30 kg/m^2^)	76 (27.84)	103 (26.01)	1.138 (0.704–1.841)	0.2648
Obese (BMI ≥ 30 kg/m^2^)	76 (27.84)	151 (38.13)	1.310 (0.815–2.104)	0.5973
Chemotherapy				
No	221 (80.95)	305 (77.02)	0.980 (0.570–1.686)	0.2227
Yes	52 (19.05)	91 (22.98)	REF	
Endocrine therapy				
No	117 (42.86)	159 (40.15)	1.056 (0.694–1.608)	0.4848
Yes	156 (57.14)	237 (59.85)	REF	
Targeted therapy				
No	188 (68.86)	224 (56.57)	1.395 (0.831–2.342)	0.0013
Yes	85 (31.14)	172 (43.43)	REF	
Radiation therapy				
No	243 (89.01)	341 (86.11)	0.929 (0.496–1.741)	0.2683
Yes	30 (10.99)	55 (13.89)	REF	
Surgery in the past 3 months				
No	250 (91.58)	345 (87.12)	1.069 (0.536–2.133)	0.0711
Yes	23 (8.42)	51 (12.88)	REF	
CCI, mean (SD)	2.90 (2.06)	3.52 (2.58)	1.020 (0.930–1.119)	0.0006
PHQ-8, mean (SD)	5.29 (4.43)	9.29 (5.17)	1.155 (1.106–1.207)	<0.0001
Total HCP visits, mean (SD)	11.08 (10.33)	12.85(11.81)	1.000 (0.981–1.018)	0.0406

Adj OR, adjusted odds ratio (from multivariable logistic regression with variables in the Table); BC, breast cancer; BMI, body mass index; CCI, Charlson Comorbidity Index; CI, confidence interval; COST, comprehensive score for financial toxicity; eBC, early breast cancer; FT, financial toxicity; HCP, health care provider; mBC, metastatic breast cancer; PHQ-8, 8-item Patient Health Questionnaire; REF, reference SD, standard deviation.

**Table 3 cancers-16-00062-t003:** Demographics and clinical characteristics associated with economic stress due to COVID-19.

Variable	CSS-SE ≤ 2 *n* = 374	CSS-SE > 2 *n* = 295	Adj OR (95% CI)	*p* Value
Age, years				
<50	153 (40.91)	126 (42.71)	0.781 (0.434–1.405)	0.4089
50 to 64	170 (45.45)	130 (44.07)	0.804 (0.458–1.410)	0.4458
≥65	51 (13.64)	39 (13.22)	REF	
Sex				
Male	3 (0.80)	4 (1.36)	1.851 (0.362–9.480)	0.5464
Female	371 (99.20)	291 (98.64)	REF	
Race				
White	329 (87.97)	232 (78.64)	REF	
Black	24 (6.42)	36 (12.20)	2.165 (1.175–3.990)	0.0133
Other	21 (5.61)	27 (9.15)	1.938 (1.005–3.736)	0.0481
Marital status				
Married/living with partner	278 (74.33)	183 (62.03)	1.225 (0.816–1.838)	0.3281
Other	96 (25.67)	112 (37.97)	REF	
Employment status				
Employed	211 (56.42)	130 (44.07)	REF	
Not employed	159 (42.51)	162 (54.92)	1.166 (0.772–1.762)	0.4662
Prefer not to answer	4 (1.07)	3 (1.02)	0.837 (0.158–4.442)	0.8345
Education status				
College degree or higher	283 (75.67)	182 (61.69)	REF	0.1963
Less than college degree	91 (24.33)	112 (37.97)	1.306 (0.871–1.956)	0.9850
Income				
≤USD 49,999	68 (18.18)	106 (35.93)	1.921 (1.020–3.617)	0.0432
USD 50,000 to USD 149,999	187 (50.00)	134 (45.42)	1.250 (0.798–1.958)	0.3298
≥USD 150,000	96 (25.67)	50 (16.95)	REF	
Prefer not to answer	23 (6.15)	5 (1.69)	0.359 (0.112–1.151)	0.0848
Insurance				
Commercial	248 (66.31)	167 (56.61)	REF	
Medicare/Medicaid	78 (20.86)	87 (29.49)	0.851 (0.498–1.454)	0.5549
Other	48 (12.83)	41 (13.90)	0.984 (0.573–1.690)	0.9544
Region				
Rural/small town	135 (36.10)	103 (34.92)	0.802 (0.557–1.154)	0.2350
Suburb/city	239 (63.90)	193 (65.08)	REF	
Out-of-pocket costs for BC in an average month				
USD 0	72 (19.25)	58 (19.66)	0.640 (0.343–1.196)	0.1618
USD 1–USD 100	110 (29.41)	80 (27.12)	0.634 (0.366–1.096)	0.1027
USD 101–USD 300	86 (22.99)	62 (21.02)	0.587 (0.338–1.020)	0.0588
USD 301–USD 500	48 (12.83)	35 (11.86)	0.634 (0.334–1.201)	0.1618
USD 501 or more	54 (14.44)	55 (18.64)	REF	
Do not know	4 (1.07)	5 (1.69)	1.016 (0.232–4.456)	0.9829
BC type				
eBC	193 (51.60)	132 (44.75)	REF	
mBC	181 (48.40)	163 (55.25)	1.051 (0.639–1.727)	0.8458
COVID-19 diagnosis				
No	337 (90.11)	262 (88.81)	0.886 (0.493–1.593)	0.6860
Not answered	5 (1.34)	3 (1.02)	0.432 (0.086–2.178)	0.3091
Yes	32 (8.56)	30 (10.17)	REF	
BMI				
Obese (BMI ≥ 30 kg/m^2^)	108 (28.88)	119 (40.34)	1.340 (0.880–2.040)	0.1719
Overweight (BMI ≥ 25 to <30 kg/m^2^)	105 (28.07)	74 (25.08)	0.974 (0.631–1.506)	0.9073
Underweight/normal weight (BMI < 25 kg/m^2^)	161 (43.05)	102 (34.58)	REF	
Chemotherapy				
No	303 (81.02)	223 (75.59)	1.251 (0.778–2.011)	0.3550
Yes	71 (18.98)	72 (24.42)	REF	
Endocrine therapy				
No	159 (42.51)	117 (39.66)	1.404 (0.967–2.040)	0.0746
Yes	215 (57.49)	178 (60.34)	REF	
Targeted therapy				
No	234 (62.57)	178 (60.34)	0.942 (0.596–1.487)	0.7967
Yes	140 (37.43)	117 (39.66)	REF	
Radiation therapy				
No	328 (87.70)	256 (86.78)	0.720 (0.414–1.253)	0.2451
Yes	46 (12.30)	39 (13.22)	REF	
Surgery in the past 3 months				
No	339 (90.64)	256 (86.78)	1.498 (0.832–2.696)	0.1777
Yes	35 (9.36)	39 (13.22)	REF	
CCI, mean (SD)	3.05 (2.22)	3.54 (2.58)	1.052 (0.972–1.138)	0.2113
PHQ-8, mean (SD)	6.43 (4.58)	9.22 (5.64)	1.090 (1.051–1.130)	<0.0001
Total HCP visit, mean (SD)	11.76 (10.26)	12.60 (12.41)	0.997 (0.981–1.012)	0.6759

Adj OR, adjusted odds ratio (from multivariate logistic regression with variables in the Table); BC, breast cancer; BMI, body mass index; CCI, Charlson Comorbidity Index; CI, confidence interval; COVID-19, coronavirus disease 2019; CSS-SE, COVID-19 stress scale–socioeconomics; eBC, early breast cancer; FT, financial toxicity; mBC, metastatic breast cancer; PHQ-8, 8-item Patient Health Questionnaire; REF, reference; SD, standard deviation.

## Data Availability

Upon request, and subject to review, Pfizer will provide the data that support the findings of this study. Subject to certain criteria, conditions and exceptions, Pfizer may also provide access to the related individual de-identified participant data. See https://www.pfizer.com/science/clinical-trials/trial-data-and-results for more information.

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
