# Peer review of "Financial Toxicity among Patients with Breast Cancer during the COVID-19 Pandemic in the United States"

_cancers, 2023, doi:10.3390/cancers16010062_

Round 1

Reviewer 1 Report

Comments and Suggestions for Authors

Esteemed Authors,

I read with great interest your article about financial burden of breast carcinoma patients during the Covid-19 pandemics.

However, there are some aspects that require your attention.

Line 12- why is so important this affiliation, at the end of the manuscript you have a section about the possible conflicts of interests generated by the authors, please remove this line

The manuscript has many abbreviations, you need to include a list of abbreviations at the end of the manuscript.

In the discussion section you need to underline that the addressability of the patients towards healthcare systems declined. Many cases postponed seeking medical advice and thus the severity of the pathology increased. For example early detection and prevention of the carcinomas declined. The general population limited the encounters with the healthcare systems. Proof to this is the very small number of cases detected with breast carcinoma during the pandemics.

Expand the subsection in which to describe future implications of the study, future developments of the study. How can we practically decrease the financial burden for breast cancer patients?

Looking forward to receiving the improved version of the manuscript.

Reviewer 2 Report

Comments and Suggestions for Authors

This is an interesting area, exploring the prevalence of financial toxicity (FT) among patient with breast cancer, specifically during the period of the COVID-19 pandemic.

The methods used in the paper are generally sound and are supported using simple (but appropriate) statistical analysis, and the overall findings are useful for policy makers.

I found the 'emphasis' of the paper to be confusing, however. Whilst it is very useful to see the rates of FT in this population, the focus on the COVID-19 pandemic seems unnecessary and potentially distracts from the main messages. I appreciate that the data collection took place during the pandemic, but the inclusion of "during the COVID-19 pandemic" in the title and main conclusion statements risks readers inferring that these conclusions are no longer relevant (they almost certainly are).

If the objective of the authors was to specifically say that FT is 'more' (or perhaps 'less') prevalent during a pandemic, then this could be outlined more clearly in the introduction, discussion and conclusion sections, and should really be supported by comparisons with similar data from non-pandemic years.

If the objective was simply to highlight the overall importance of FT in BC patients, then I do not see why COVID-19 needs to be mentioned in the title. It would be better to present the paper as an analysis of FT in BC, and then show the results of the CSS-SE groups as sub-analysis, also highlighting in the discussion section that the results may be less generalisable because they were collected using data from an unusual time.

It would also be helpful to update the paper's title to be clear that the research and data collection were conducted in the United States.

Overall, the paper is excellent, but would benefit from additional clarity around its focus.
